# Retinoic Acid-Mediated Inhibition of Mouse Coronavirus Replication Is Dependent on IRF3 and CaMKK

**DOI:** 10.3390/v16010140

**Published:** 2024-01-18

**Authors:** Justin H. Franco, Ryan A. Harris, William G. Ryan, Roger Travis Taylor, Robert E. McCullumsmith, Saurabh Chattopadhyay, Zhixing K. Pan

**Affiliations:** 1Department of Medical Microbiology and Immunology, University of Toledo College of Medicine and Life Sciences, Toledo, OH 43614, USAsaurabh.chattopadhyay@uky.edu (S.C.); 2Department of Neurosciences and Neurological Disorders, University of Toledo College of Medicine and Life Sciences, Toledo, OH 43614, USA; 3Department of Microbiology Immunology and Molecular Genetics, University of Kentucky College of Medicine, Lexington, KY 40536, USA

**Keywords:** retinoic acid, coronavirus, MHV, IRF3, CaMKK

## Abstract

The ongoing COVID-19 pandemic has revealed the shortfalls in our understanding of how to treat coronavirus infections. With almost 7 million case fatalities of COVID-19 globally, the catalog of FDA-approved antiviral therapeutics is limited compared to other medications, such as antibiotics. All-trans retinoic acid (RA), or activated vitamin A, has been studied as a potential therapeutic against coronavirus infection because of its antiviral properties. Due to its impact on different signaling pathways, RA’s mechanism of action during coronavirus infection has not been thoroughly described. To determine RA’s mechanism of action, we examined its effect against a mouse coronavirus, mouse hepatitis virus strain A59 (MHV). We demonstrated that RA significantly decreased viral titers in infected mouse L929 fibroblasts and RAW 264.7 macrophages. The reduced viral titers were associated with a corresponding decrease in MHV nucleocapsid protein expression. Using interferon regulatory factor 3 (IRF3) knockout RAW 264.7 cells, we demonstrated that RA-induced suppression of MHV required IRF3 activity. RNA-seq analysis of wildtype and IRF3 knockout RAW cells showed that RA upregulated calcium/calmodulin (CaM) signaling proteins, such as CaM kinase kinase 1 (CaMKK1). When treated with a CaMKK inhibitor, RA was unable to upregulate IRF activation during MHV infection. In conclusion, our results demonstrate that RA-induced protection against coronavirus infection depends on IRF3 and CaMKK.

## 1. Introduction

First emerging in 2019, severe acute respiratory syndrome coronavirus 2 (SARS-CoV-2) has infected over 700 million people globally and caused approximately 7 million deaths [1,2]. As the causative agent of coronavirus disease 2019 (COVID-19), SARS-CoV-2 is the latest novel coronavirus to appear within the last twenty years [1,2]. Similar to other pathogenic coronaviruses (e.g., SARS-CoV and MERS-CoV), SARS-CoV-2 primarily causes viral pneumonia that can manifest as a mild flu-like condition or as acute respiratory distress syndrome (ARDS) [2,3,4]. Excessive viral replication in lung tissue leads to a dysregulated proinflammatory immune response that results in ARDS and death from respiratory failure [3,4]. The excessive release of proinflammatory cytokines also promotes widespread clotting that can obstruct blood flow and precipitate organ failure [3,4].

Typically, viral infections are cleared by the innate and adaptive immune system. While the adaptive immune system provides protection against repeat infection, the innate immune system is crucial for controlling early infection, especially against novel pathogens [5]. The innate immune system’s primary antiviral mechanism is the type-I interferon (IFN) response, which relies on pattern recognition receptors (PRRs) to activate IFN regulatory factor 3 (IRF3) [6,7,8]. IRF3 activation results in IFNβ secretion and IFN-stimulated gene (ISG) expression, which leads to antiviral protein production [6,7,9,10]. Antiviral proteins, such as IFN-induced tetratricopeptide repeat proteins (Ifit), play a crucial role in disrupting viral replication [5]. Unfortunately, coronaviruses express numerous proteins to inhibit the type-I IFN response [4,8,11,12]. A well-characterized example of a potent type-I IFN response antagonist is the coronavirus nucleocapsid (N) protein [11]. In coronaviruses, N protein interferes with the type-I IFN response by inhibiting IRF3 activation and by targeting retinoic acid-inducible gene-I (RIG-I), which is a key intracellular PRR for viral RNA [11,12,13,14]. Similar N protein activity is also seen in model animal coronaviruses, such as mouse hepatitis virus strain A59 (MHV), where it inhibits RIG-I activation and antagonizes antiviral proteins [15,16].

Because coronaviruses effectively block the type-I IFN response, much research has been devoted to investigating pharmaceutical treatments. While vaccinations have successfully decreased COVID-19 mortality rates, treatments for severe disease are limited [17,18]. Currently, Remdesivir and Paxlovid are the only antivirals approved by the US FDA for mild-to-moderate COVID-19 [18,19]. Other therapeutics, such as Molnupiravir, have also achieved emergency use authorization from the US FDA for moderate disease [18]. However, no antivirals have yet been approved for severe COVID-19 [18]. 

A potential therapeutic for COVID-19 that has garnered much interest is activated vitamin A, or all-trans retinoic acid (RA). Although predominantly used to treat acute promyelocytic leukemia and inflammatory skin conditions (e.g., psoriasis and acne vulgaris), RA also exhibits antiviral properties [20,21]. Previous in vitro studies have shown that RA confers protection against viral infection by promoting the type-I IFN response [22,23,24]. RA has also demonstrated direct antiviral activity, functioning as an antagonist against different viral proteins [25,26]. Although RA has been shown to decrease coronavirus replication in vitro, by interfering with viral proteases and lipid synthesis, its impact on the type-I IFN response has yet to be fully described [26,27]. In the present study, we show that RA-induced activation of the type-I IFN response is dependent on IRF3 and Calcium/calmodulin kinase kinase (CaMKK) activity.

## 2. Materials and Methods

### 2.1. Cultured Cells and Virus

Each cell line used in the study was maintained in the author’s laboratory. Mouse L929 fibroblast cells were purchased from ATCC and cultured in Dulbecco’s Modified Eagle’s Medium (DMEM) supplemented with 10% fetal bovine serum (FBS) and 1% penicillin–streptomycin. Mouse RAW 264.7-Lucia and RAW 264.7-Lucia-KO-IRF3 macrophages, purchased from InvivoGen (San Diego, CA, USA), were cultured in DMEM containing 10% FBS, 1% penicillin-streptomycin, and 0.1% Normocin. 

Mouse hepatitis virus strain A59 (MHV) was obtained from ATCC (VR-764). Viral stocks were maintained by propagating MHV in our mouse L929 fibroblasts. Virus titers were quantified by performing a plaque assay in L929 cells overlayed with 1% carboxy-methyl cellulose (CMC). 

### 2.2. Reagents

An RNA purification kit (i.e., Invitrogen PureLink RNA Mini Kit), commercial Western blot gel (i.e., NuPage 4–12% BT Gel), polyvinylidene difluoride (PVDF) membrane, gel running buffer (i.e., BOLT MES SDS Running buffer 20×), gel transfer buffer (i.e., BOLT Transfer buffer 20×), stripping buffer (i.e., Restore Plus Western Blot stripping buffer), loading buffer (i.e., BOLT LDS sample buffer 4×), chemiluminescence kit (i.e., Pierce ECL Western Blotting Substrate), Tali Dead Cell red reagent, Dulbecco’s phosphate-buffered saline (DPBS), DMEM, and FBS were purchased from ThermoFisher Scientific (Waltham, MA USA). Dimethyl sulfoxide (DMSO), All-trans retinoic acid (RA), Calcium/Calmodulin kinase kinase inhibitor (i.e., STO-609), and bovine serum albumin (BSA) were obtained from Sigma Aldrich (St. Louis, MO, USA). RA stock solution (0.1 M) was prepared by diluting RA in DMSO. Recombinant mouse Interferon β (IFNβ) and Poly-I:C were purchased from Bio-techne (Minneapolis, MN, USA).

### 2.3. RA Cell Survival Quantification

Mouse L929, RAW 264.7-Lucia, and RAW 264.7-Lucia-KO-IRF3 cells were grown in an incubator at 37 °C in 5% CO_2_. Cell lines were transferred to a 24-well plate at a concentration of 2 × 10^5^ cells/well. After 1 h to allow for plate adherence, cells were treated with 500 µL of different RA concentrations (e.g., 0.1 µM, 1 µM, 10 µM, and 100 µM). Treated cells were left in the incubator for 10 h before being washed with DPBS. Cells were then administered 300 µL of Trypsin and transferred to a 1.5 mL Eppendorf tube for analysis. Then, 100 µL of each sample was given 1 µL of Tali Dead Cell red reagent and left to incubate for 5 min. After incubation, cell death was analyzed using a Tali-based image cytometer.

### 2.4. MHV Infection

Mouse L929, RAW 264.7-Lucia, and RAW 264.7-Lucia-KO-IRF3 cells were grown in an incubator at 37 °C in 5% CO_2_. Cell lines were transferred to a 24-well plate and given 1 h to adhere. Once plated, cells were maintained in DMEM supplemented with 10% FBS and 1% penicillin–streptomycin. After adhering, positive control cells were administered either 500 µL of 1000 IU/mL IFNβ or 10 µg/mL Poly-I:C for 16 h. Following IFNβ and Poly-I:C pretreatment, cells were infected with MHV that was diluted in supplemented DMEM. MHV infections were conducted using a multiplicity of infection (MOI) of 1 or 3. Cells were incubated in MHV media for 1 h, when the well plate was gently tilted side to side every 20 min. MHV media was then removed, and infected cells were washed with DPBS before replacing with supplemented DMEM. After incubating for 1 h, cells were treated with 1 µM, 10 µM RA, or 0.1% DMSO. Depending on the experiment, cells were left to incubate for either 4, 9, 19, or 23 h (i.e., total incubation times of 5, 10, or 20 h). 

### 2.5. Viral Plaque Assay

Cellular supernatant was collected from all cells after completing the infection study. Collected supernatant was centrifuged at 1400× *g* for 10 min in a 5 °C fridge to remove cellular debris. The supernatant from each sample was diluted in DMEM supplemented with 10% FBS and 1% penicillin–streptomycin. Sample supernatant was prepared in the following dilutions: 10^−2^, 10^−3^, and 10^−4^. L929 fibroblasts were seeded into a 24-well plate, with 2 × 10^5^ cells/well. Cells were grown in supplemented DMEM and given 1 h to adhere. After adhering, cells were infected with diluted sample supernatant for 1 h. During infection, the plate was gently tilted side to side every 20 min. After infection, the diluted supernatant was removed, and cells were washed with DPBS. DPBS media were then replaced with 1% CMC overlay, and the cells were left to incubate for approximately 48 h. Following incubation for 48 h, the CMC overlay was replaced with 10% formalin solution. After 30 mins of incubation, the formalin solution was removed, and the cells were washed gently with water. Cells were then administered 1% crystal violet solution, diluted in 25% ethanol, for 1 h. Next, the crystal violet solution was washed off with water, and the plate was left to dry overnight. Once dry, the number of plaques was counted, and the viral titer was calculated in plaque-forming units per ml (PFU/mL). 

### 2.6. RNA Extraction and qRT-PCR

Following the MHV infection study, cellular RNA was extracted using TRIzol reagent purchased from ThermoFisher Scientific (Waltham, MA USA). Collected RNA was precipitated according to the instructions of the TRIzol kit. The total RNA concentration was then quantified using a ThermoFisher Scientific NanoDrop 2000 spectrophotometer. cDNA was prepared using 1000 ng RNA samples and the Superscript IV Vilo kit obtained from ThermoFisher Scientific. Sample cDNA was generated after undergoing standard heat cycles (i.e., 25 °C for 10 min, 50 °C for 15 min, and 85 °C for 5 min). Sample cDNA was prepared for qRT-PCR analysis using PowerUP SYBR green master mix, purchased from ThermoFisher Scientific. The preparation was analyzed with a Roche Light Cyler 96 system, with 15 µL/well in triplicate. Primers for Ifit1, Ifit3, MHV nucleocapsid (MVH-N), β-Actin, and 18 s rRNA were prepared by Integrated DNA Technologies Inc. (Coralville, IA, USA) (Table 1). β-Actin and 18s rRNA were used to normalize gene expression for L929 fibroblasts and RAW 264.7 macrophages, respectively. Data generated from the Roche Light Cycler 96 software underwent statistical analysis using GraphPad Prism 7.

### 2.7. Protein Extraction and Western Blot Analysis

Protein was extracted from cell samples using M-PER mammalian protein extraction reagent, purchased from ThermoFisher Scientific. M-PER lysis buffer was supplemented with HALT Protease & Phosphatase single-use inhibitor cocktail. After extraction, protein concentration was determined using a Pierce Bicinchoninic acid (BCA) Protein Assay kit. Protein concentrations were measured by mixing 5 µL sample with 5 µL M-PER buffer in each well, along with 200 µL/well of mixed BCA reagents A and B (i.e., A and B mixed at a ratio of 50:1). All reagents were combined according to kit instructions, and absorbance was measured using a BMG Labtech (Cary, NC, USA) FLUOstar Omega plate reader.

Then, 20 µg protein samples were mixed with 10 µL BOLT LDS sample buffer 4× and heated at 90 °C for 10 min to ensure protein denaturation. With a final volume of 40 µL, each protein sample was loaded onto a 12-well NuPage 4–12% Bis-Tris gel and electrophoresed in BOLT MES SDS Running buffer 1× for 32 min (i.e., 100 V for 10 min, followed by 200 V for 22 min). Transfer to PVDF membrane was conducted at 17 V for 1 h. Blots were incubated in 5% BSA blocking buffer for 1 h before primary antibody incubation. Primary antibodies for MHV-N and mouse β-Actin were acquired from Rockland (Limerick, PA, USA) and ThermoFisher Scientific, respectively. Using 2% BSA, MHV-N antibody was diluted to a concentration of 1/5000, while mouse β-Actin antibody was prepared at a concentration of 1/3000. Blots were allowed to incubate in primary antibody solution for at least 2 h before being washed with TBST buffer 1×. After washing, blots were incubated in secondary antibody (e.g., anti-rabbit or anti-mouse IgG) for 1 h at a concentration of 1/3000. Using a Pierce ECL Western Blotting Substrate kit, the chemiluminescence of each blot was measured with a G-Box imaging system. Differences in band intensity were quantified using Image J (Version 1.53a) for densitometric analysis. Statistical analysis was then conducted using GraphPad Prism 7, with experiments performed in triplicate. Blots were treated with Restore Plus Western Blot stripping buffer for 15 min and washed with TBST 1X. After incubating in 5% BSA blocking buffer for 30 min, primary antibodies were reapplied to examine other proteins of interest.

### 2.8. IFN Regulatory Factor Activation Assay

RAW-Lucia and RAW-Lucia-KO-IRF3 cells were seeded into a 24-well plate at 2 × 10^5^ cells/well. Cells were pretreated for 2 h with supplemented DMEM, 4000 IU/mL IFNβ, or 10 µM STO-609. Pretreatment media were then removed, and cells were infected with MHV for 1 h at an MOI of 3. Following infection, cells were washed with DPBS and incubated in supplemented DMEM, 4000 IU/mL IFNβ, or 10 µM STO-609. Then, 1 h after MHV media removal, 1 µM and 10 µM RA were added to the appropriate cells. Sample supernatant was collected 19 h later (i.e., total incubation of 20 h). Collected supernatant from RAW 264.7-Lucia and RAW 264.7-Lucia-KO-IRF3 cells was examined in an Interferon regulatory factor (IRF) activation assay. Then, 20 µL of sample supernatant was added to a 96-well plate in quadruplicate. Using InvivoGen’s Quanti-Luc reagent, 50 µL of reagent was added to each well. Fluorescence was measured with a FLUOstar omega plate reader according to assay instructions. Statistical analysis was then conducted using GraphPad Prism 7.

### 2.9. RNA Extraction and mRNA-Sequencing Analysis

RNA was extracted using TRIzol reagent and isolated with the Invitrogen PureLink RNA Mini Kit. Extracted RNA was isolated according to Invitrogen’s kit instructions. Total RNA concentration was measured using the NanoDrop 2000 system. Afterward, RNA samples were shipped to NovoGene for mRNA sequencing. Raw data generated by NovoGene were then analyzed using R software (Version 4.2.2).

### 2.10. Differential Gene Expression Analysis

Transcript abundances were quantified from paired-end reads (raw FASTQ files) against the Mus musculus reference transcriptome (GRCm39) using kallisto (version 0.43.1) [28]. Transcriptome-wide gene counts were obtained through gene-level transcript summarization with the tximport R package (version 1.26.1) [29]. Differentially expressed gene (DEG) analysis was performed after filtering for low-expression genes with the filterByExpr function using the edgeR package (version 3.40.2) [30].

### 2.11. Heatmap and Volcano Plot Generation

Expression profiles of the top thousand most variable genes (defined by greatest sample variance of the scaled and centered log-transformed TMM-normalized counts per million) were visualized using the ComplexHeatmap R package (version 2.14.0) [31]. Differential expression profiles were visualized using standard volcano plots. The difference in gene expression (log2FoldChange) between experimental comparisons was plotted against significance values (−log10pvalue). Genes showing the greatest change (top 10 by abs. log2FoldChange) were labeled. Dashed line indicates significance threshold of *p* < 0.05.

### 2.12. Statistical Analysis

Densitometric analysis of Western blot band intensity was conducted using Image J.

Each statistical analysis was conducted using GraphPad Prism 7 software and Microsoft (Redmond, WA, USA) Excel. *p* values were calculated using Dunnett’s One-Way ANOVA, with values less than 0.05 defined as statistically significant. Figures with associated *p* values are representative of at least three independent experiments. 

## 3. Results

### 3.1. RA Confers Protection against MHV Infection

We used MHV to investigate RA’s antiviral properties during coronavirus infection [32,33,34,35]. Similar to human coronaviruses, MHV can inhibit the type-I IFN response and infect lung tissue [15,16,36,37]. In our study, we examined mouse fibroblast and macrophage cell lines because they are commonly present in lung tissue and mediate pulmonary inflammation [36,38]. Mouse L929 fibroblasts were infected with MHV at a multiplicity of infection (MOI) of 0.1. High-dose RA (i.e., 100 µM) was administered 1 h after MHV infection for a total incubation time of 6. Using harvested cellular mRNA from MHV-infected L929 cells, we performed a qRT-PCR analysis. The results demonstrated that high-dose RA upregulated Ifit1, Ifit3, and IFNβ mRNA expression during MHV infection at MOI 0.1 (Figure 1A–C). When examined at a higher infectious dose and longer total incubation time (i.e., MOI 1 and 10 h), RA-treated L929 cells exhibited decreased MHV-N mRNA and increased Ifit1 and Ifit3 mRNA expression (Figure 1D–F). A total of 100 µM RA was used initially because of its potent effect on gene transcription [39]. Likewise, RA is well tolerated by L929 fibroblasts and exhibits negligible cell toxicity (Appendix A).

A plaque assay was performed to determine if RA-induced reductions of MHV-N mRNA were associated with decreased viral titers. The supernatant was harvested from L929 cells infected with MHV at an MOI of 1 and treated with varying concentrations of RA. The results demonstrated that RA significantly reduced MHV titers at concentrations of 1 µM, 10 µM, and 100 µM (Figure 2A,B). Only 0.1 of µM RA was unable to reduce MHV replication in L929 cells. 

### 3.2. IRF3 Is Required for RA-Induced MHV Suppression

While coronaviruses are adept at downregulating the type-I IFN response, RA treatment can overcome this activity to confer protection against infection. In previous studies, RA’s antiviral effect has been demonstrated to depend on the early stages of the type-I IFN response, namely, the upregulation of RIG-I and IFNβ [22,23,24]. As a key transcription factor, IRF3 activation by RIG-I causes induction of IFNβ [5,6,7]. To investigate whether RA’s antiviral capacity directly depends on IRF3, we examined its effect in mouse RAW 264.7-Lucia (WT) and RAW 264.7-Lucia-KO-IRF3 (IRF3-KO) macrophage reporter cell lines. Similar to our L929 fibroblasts, RA is well tolerated in our WT and IF3-KO cell lines and exhibits negligible cell toxicity (Appendix A).

A qRT-PCR analysis of MHV-N mRNA expression was performed using MHV-infected WT and IRF3-KO cells at an MOI of 1. A 16 h pretreatment of 1000 IU/mL IFNβ and 10 µg/mL Poly-I:C was used as a positive control. Both IFNβ and Poly-I:C are potent activators of the type-I IFN response, stimulating IFNβ and PRRs, respectively. Because lower RA concentrations (i.e., 1 µM and 10 µM) were sufficient to suppress MHV replication, the following experiments were performed at concentrations under 100 µM. 

The analysis showed that RA downregulated MHV-N mRNA expression in WT cells but not in IRF3-KO cells (Figure 3A). Surprisingly, RA-induced upregulation of Ifit1 and Ifit3 was only seen in the IRF3-KO cells (Figure 3B,C). However, the levels of Ifit expression were lower than those seen in the WT cells. As expected, Poly-I:C-mediated inhibition of MHV was also disrupted in IRF3-KO cells, while IFNβ was still effective in inhibiting MHV replication in both WT and IRF3-KO cells. The results indicate that RA triggers the induction of antiviral Ifit genes via IRF3′s transcriptional activity. We noted that IRF3-KO cells did not completely abolish Ifit expression. However, this is not unexpected because other IRFs (e.g., IRF7 and IRF5) can also induce these genes.

To determine if RA on MHV-N mRNA expression was associated with reduced viral titers, a Western blot and plaque assay were accomplished. Western blot analysis of MHV-N protein revealed that RA reduced MHV-N expression in WT cells infected at an MOI of 3 (Figure 4A–C). Also, WT cells infected at an MOI of 1 exhibited lower MHV titers following RA treatment (Figure 4D–F). Unlike the WT cells, the IRF3-KO exhibited no benefit from RA treatment following MHV infection (Figure 4A–F). These findings illustrate that RA-induced suppression of MHV replication is dependent on IRF3.

### 3.3. RA Promotes Activation of the Type-I IFN Response via IRF3 and CaMKK

Our previous experiments demonstrate that RA suppressed MHV replication in mouse L929 and WT cells. Protection against infection was associated with reduced MHV-N mRNA and protein. Since RA is associated with upstream activation of the type-I IFN response (i.e., increased IFNβ secretion), we investigated whether RA could induce activation of the type-I IFN response [22,23,24]. For this purpose, we conducted an IRF luciferase activation assay using WT and IRF3-KO RAW reporter cells infected at an MOI of 3 for a total incubation time of 20 h. Both cell lines express an IRF-inducible Lucia luciferase reporter system, thus enabling quantification of type-I IFN activation. 

In our assay, RA increased luciferase expression in WT cells, thus signifying increased IRF activation (Figure 5A). As expected, only 4000 IU/mL IFNβ was able to increase IRF activation in IRF3-KO cells. Due to the absence of IRF3, other IRFs (e.g., IRF7, IRF9, etc.) become the main activator of the type-I IFN response and induce luciferase secretion. These results further support that RA-induced type-I IFN responses depend on IRF3 and its transcriptional activity.

Because of RA’s multifaceted influence on different signaling pathways, we performed an mRNA-sequencing analysis of our WT and IRF3-KO cells. For our experiment, cells were infected with MHV at an MOI of 1 and allowed to incubate for a total of 10 h. RA treatment resulted in widespread changes in gene expression in both WT and IRF3-KO cells (Figure 6A–E and Figure 7A–E). Indeed, RA treatment caused robust induction of genes in both uninfected and MHV-infected cells. Of the genes that were upregulated by RA, the most unexpected was CaMKK1.

CaMKKs (e.g., CaMKK1 and CaMKK2) are major signaling proteins in the CaM pathway, activating important downstream CaM proteins, such as CaMKI and CaMKIV [40,41,42,43]. Although CaMKK activation of downstream CaM proteins is classically associated with metabolism, recent studies have demonstrated its positive effect on macrophage activation and immune function [44]. To validate our RNA-sequencing analysis, we examined CaMKK’s influence on RA-induced IRF activation during MHV infection. Using 10 µM STO-609 (i.e., CaMKK inhibitor), we repeated the IRF activation assay study. Treatment with STO-609 blocked RA’s ability to upregulate IRF activation during MHV infection (Figure 5B). 

To verify that STO-609 suppression of RA activity was associated with increased viral titers, we conducted a plaque assay using cellular supernatant from the IRF activation assay. WT cells treated with STO-609 and RA exhibited similar MHV titers to untreated infected controls (Figure 5C,D). Therefore, RA-mediated activation of the type-I IFN response is dependent on IRF3 and CamKK activity.

## 4. Discussion

RA is a versatile compound with anti-inflammatory and antiviral properties [21,22,23,24,45]. During viral infection, RA promotes activation of the type-I IFN response and stimulates ISG expression [22,23,24]. Using higher viral inoculums (MOI 1 and 3) than previous studies (e.g., MOI 0.05 to 0.5), we showed that RA confers protection against MHV infection [22,23,24]. This protection was associated with decreased viral titers, reduced expression of MHV-N, and increased IRF activation. RA’s antiviral effect against MHV infection was dependent on IRF3. Without IRF3, RA treatment failed to suppress MHV replication. However, RA was still able to upregulate low levels of Ifit mRNA expression in IRF3-KO cells, likely due to the activation of other IRFs by RA [5,46]. Ifit expression is especially important during MHV infection, where it is required for effective viral clearance [47,48]. 

Our mRNA sequencing analysis revealed that RA increased expression of CaMKK1 (CaMKKα), which is an isoform of CaMKK2 (CaMKKβ). CaMKKs are key signaling molecules that are responsible for phosphorylating downstream proteins, such as CaMKI, CaMKIV, and AMPK [42,43,49,50]. While each CaMKK target is generally involved in regulating cell metabolism, AMPK has been shown to also contribute to antiviral defense [40,51]. AMPK can induce the type-I IFN response by activating STING, which plays a role in phosphorylating IRF3 [51,52]. Disruption of CaMKK activity may interrupt cellular IRF3 phosphorylation pathways, thus reducing RA’s capacity to protect against coronavirus infection. Along with phosphorylating AMPK, CaMKK regulates macrophage function by inducing type II activation (i.e., M2b macrophage) [44,53]. M2b macrophages regulate immune activity by secreting proinflammatory (e.g., TNFα, IL-6, etc.) and anti-inflammatory (e.g., IL-10) cytokines [53]. RA-induced upregulation of CaMKK1 may prevent an excessive inflammatory response from taking place by promoting M2b activity. While previous studies have demonstrated RA’s capacity to increase calcium uptake, our work is the first to demonstrate a direct role of CaMKK in RA’s antiviral properties [54].

RA-induced CaMKK1 expression may also contribute to RA’s psychiatric side effect profile. Although an uncommon side effect, vitamin A derivatives are associated with depression and psychosis [55,56]. Their effect on brain activity is attributed to their ability to regulate gene expression in the hypothalamus [55,56]. CaM signaling proteins, such as CaMKIV and CaMKK1, also play an important role in the brain by regulating mood and long-term memory [57,58]. Whether RA’s psychiatric side effects are mediated by CaMKK is an avenue of investigation warranting further study.

Although our results show that RA-mediated suppression of coronavirus infection is dependent on IRF3 and CaMKK activity, this study has several limitations. Early qRT-PCR studies only examined the effect of high doses of RA in mouse L929 cells infected with MHV. Later studies using WT cells demonstrated that lower RA doses could also reduce MHV-N mRNA expression. However, RAW macrophages are less affected by MHV infection than L929 fibroblasts, as evidenced by their lower overall MHV titers. Future studies examining RA’s mechanism of action during coronavirus infection should investigate more susceptible cell types, such as lung epithelial cells. Likewise, the use of other knockout cell lines (e.g., IRF7-KO, IRF9-KO, etc.) and more extensive mRNA sequencing can further characterize RA’s multifaceted antiviral activity. 

## 5. Conclusions

Our results illustrate that RA confers protection against coronavirus infection by stimulating IRF activation. RA-mediated suppression of coronavirus replication is dependent on IRF3 and CaMKK activity. Although further mechanistic studies are needed to define the role of CaM signaling during viral infection, its effect on RA’s antiviral activity offers an important insight into understanding RA’s complex mechanism of action.

## Figures and Tables

**Figure 1 viruses-16-00140-f001:**
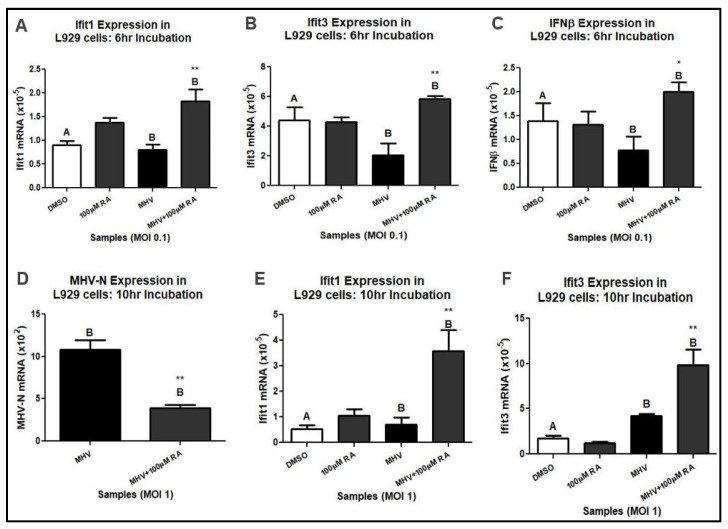
A total of 100 µM RA upregulates antiviral mRNA expression. In L929 cells infected with MHV at MOI 0.1, RA increased (**A**) Ifit1, (**B**) Ifit2, (**C**), and IFNβ mRNA expression. When administered to cells infected with MHV at MOI 1, RA decreased (**D**) MHV-N mRNA and increased (**E**) Ifit1 and (**F**) Ifit3 expression. Data were taken from experiments performed in triplicate. Statistical significance was calculated by Dunnett’s One-Way ANOVA, with comparisons to DMSO labeled as “A” and comparisons to MHV labeled as “B”. * *p* < 0.05, ** *p* < 0.01.

**Figure 2 viruses-16-00140-f002:**
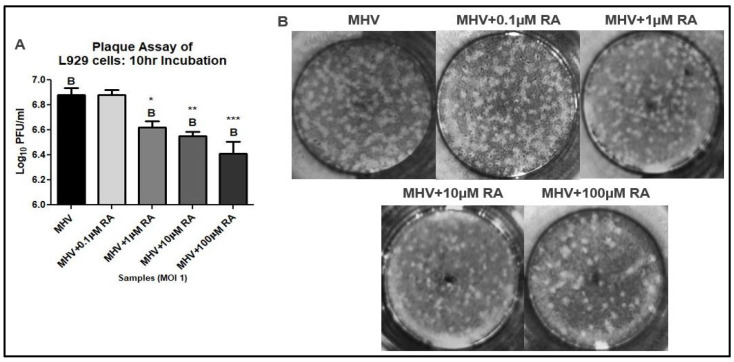
RA reduces MHV replication in L929 cells. (**A**) RA reduced MHV titers at all concentrations (i.e., 1 µM, 10 µM, and 100 µM), except 0.1 µM. (**B**) Representative plaques demonstrate decreased plaque formation at RA concentrations above 0.1 µM. Data were taken from experiments performed in triplicate. Statistical significance was calculated by Dunnett’s One-Way ANOVA, with comparisons to MHV labeled as “B”. * *p* < 0.05, ** *p* < 0.01, *** *p* < 0.001.

**Figure 3 viruses-16-00140-f003:**
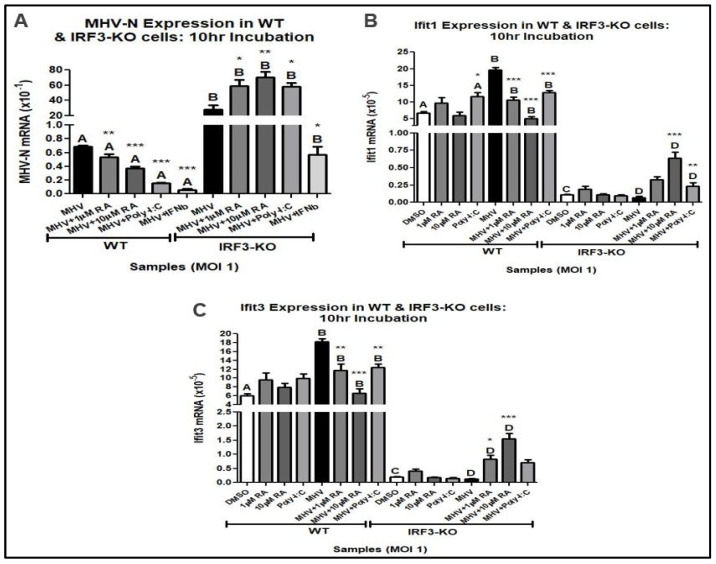
A total of 1 µM and 10 µM RA upregulates antiviral mRNA expression. (**A**) RA downregulates MHV-N mRNA expression in WT cells but not in IRF3-KO cells. IRF3-KO cells exhibited increased expression of (**B**) Ifit1 and (**C**) Ifit3 when treated with RA. Data were taken from experiments performed in triplicate. Statistical analysis was calculated by Dunnett’s One-Way ANOVA, with comparisons to WT-DMSO labeled as “A”, comparisons to WT-MHV labeled as “B”, comparisons to IRF3-KO-DMSO labeled as “C”, and comparisons to IRF3-KO-MHV labeled as “D”. * *p* < 0.05, ** *p* < 0.01, *** *p* < 0.001.

**Figure 4 viruses-16-00140-f004:**
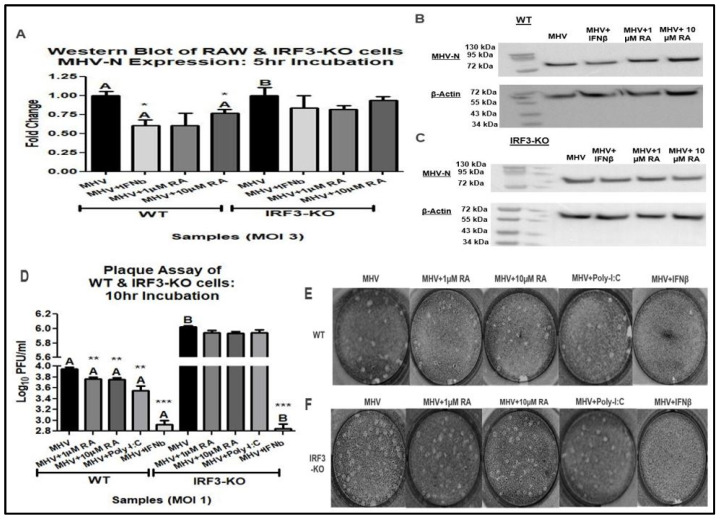
RA-induced suppression of MHV is IRF3-dependent. (**A**–**C**) A total of 1 µM RA significantly reduced MHV-N protein expression in WT cells but not IRF3-KO cells. (**D**) RA decreased MHV titers in WT cells and exhibited no effect in IRF3-KO cells. (**E**,**F**) Representative plaques demonstrate that RA reduced MHV replication in infected WT cells. Data were taken from experiments performed in triplicate. Statistical analysis was calculated by Dunnett’s One-Way ANOVA, with comparisons to WT-MHV labeled as “A” and comparisons to IRF3-KO-MHV labeled as “B”. * *p* < 0.05, ** *p* < 0.01, *** *p* < 0.001.

**Figure 5 viruses-16-00140-f005:**
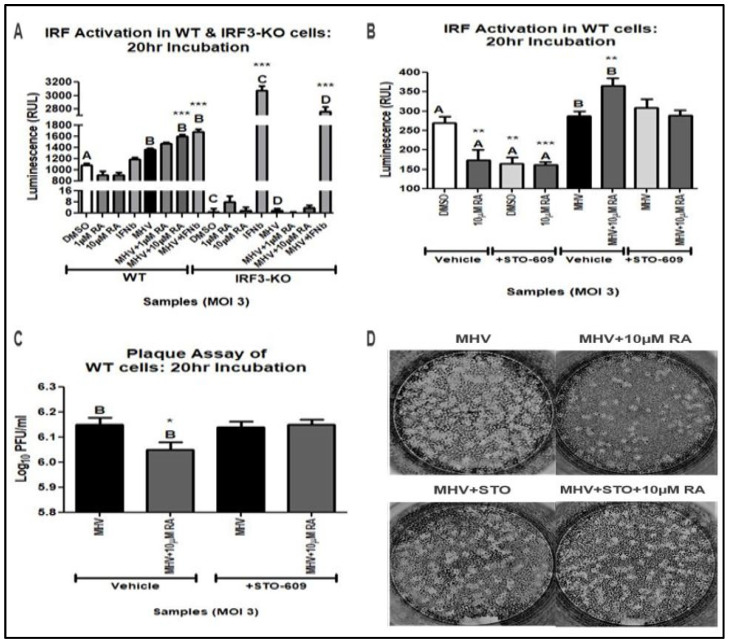
RA-mediated MHV suppression is dependent on IRF3 and CaMKK. (**A**) A total of 10 µM RA induced IRF activation in infected WT cells but not IRF3-KO cells. In IRF3-KO cells, only 4000 IU/mL of IFNβ could induce IRF activation. STO-609 (STO) disrupted RA’s ability to induce (**B**) IRF activation and reduce (**C**) MHV titers in WT cells. (**D**) Representative plaques demonstrate that STO disrupts RA’s antiviral activity. Data were taken from experiments performed in triplicate and quadruplicate. Statistical significance was calculated by Dunnett’s One-Way ANOVA, with comparisons to WT-DMSO labeled as “A”, comparisons to WT-MHV labeled as “B”, comparisons to IRF3-KO-DMSO labeled as “C”, and comparisons to IRF3-KO-MHV labeled as “D”. * *p* < 0.05, ** *p* < 0.01, *** *p* < 0.001.

**Figure 6 viruses-16-00140-f006:**
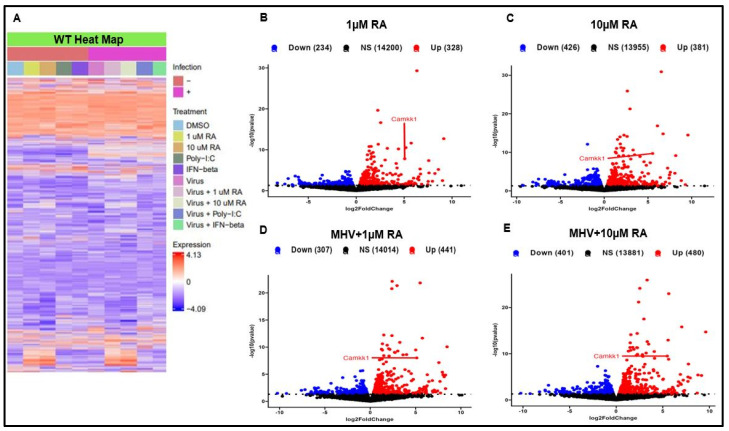
RA induces widespread gene expression in WT cells. (**A**) Heatmap illustrates differential gene expression between uninfected and infected WT cells. Volcano plots show that in both (**B**,**C**) uninfected and (**D**,**E**) infected WT cells, RA upregulates CaMKK1 expression. Data were taken from experiment performed in singlicate.

**Figure 7 viruses-16-00140-f007:**
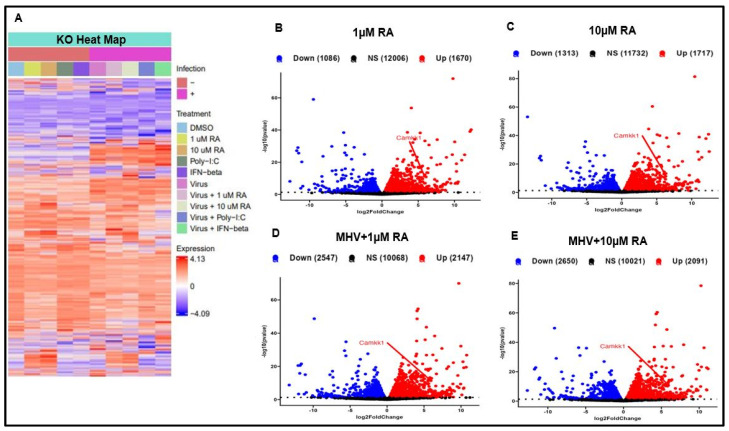
RA induces widespread gene expression in IRF3-KO cells. (**A**) Heatmap illustrates differential gene expression between uninfected and infected IRF3-KO cells. Volcano plots show that in both (**B**,**C**) uninfected and (**D**,**E**) infected IRF3-KO cells, RA upregulates CaMKK1 expression. Data were taken from experiment performed in singlicate.

**Table 1 viruses-16-00140-t001:** DNA primers used during qRT-PCR analysis.

Gene	Primer Sequence	
	Forward	Reverse
Ifit1	TTT CCG TAG GAA ACA TCG CGT	TGC TTG TAG CAG AGC CCT TTT
Ifit2	AGT ACA ACG AGT AAG GAG TCA CT	AGG CCA GTA TGT TGC ACA TGG
Ifit3	GAA GCT GAA GGG GAG CGA TT	TCC CTG TAA CGG CAC ATG AC
MHV-N	GGA CAG GGA GTG CCT ATT GC	TGG GGC CCT GTT CCA AGA TA
β-Actin	GGC TT ATT CCC CTC ATT GC	CCA GTT GGT AAC AAT GCC ATG T
18s rRNA	CCG CGG TTC TAT TTT GTT GGT	CTC TAG CGG CGC AAT ACG A

## Data Availability

Data are contained within the article and Appendix A.

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
