# Peer review of "Retinoic Acid-Mediated Inhibition of Mouse Coronavirus Replication Is Dependent on IRF3 and CaMKK"

_viruses, 2024, doi:10.3390/v16010140_

Round 1
Reviewer 1 Report
Comments and Suggestions for Authors
Franco et al present an interesting study of the mechanism through which all trans retinoic acid (RA) inhibits the coronavirus mouse hepatitis virus A59 (MHV) in a mouse in vitro fibroblast (L929) and macrophage (RAW 264.7) model.
The authors provide clear rationale for this study:
effective treatments for coronaviruses remain limited and
it is known that RA inhibits some coronaviruses, but viral subversion of this inhibition can occur so identification of the downstream effectors of RA is important as these might be attractive targets for subversion-resistant antiviral design.
The authors describe their methods in helpful, clear detail and systematically show that
in L929 fibroblast cell line, RA + MHV together induce IFIT1,3, and IFNb and RA suppresses MHV RNA and MHV titer as measured in plaque assay, without substantial toxicity.
in RAW 264.7 Lucia luciferase reporter macrophage cell lines, they compare WT vs IRF3 knockout cells MHV permissivity and RA sensitivity, and find that IRF kos are roughly 1-2 log more permissive (by PCR, with MOI 1) and in IRF ko are RA-mediated MHV suppression is lost, compared to WTs. The same finding is observed measuring viral protein through western and live virus through plaque assay, in contrast to preservation of IFNb antiviral activity in IRF3 ko. It is notable that the fold change of MHV in RA treatment measured by WB and plaque assay is small but found to be significant.
The authors then used RNAseq to further clarify the mechanism of RA activity, comparing RAW wt v IRF3 ko, finding multiple differentially expressed genes, including CAMK2 upregulation by RA. To determine whether CAMK2 had a functional role in RA to IRF activation, the authors then tested impact of CAMK2 inhibitor STO 609 on RA suppression of MHV using plaque assay, and found that STO 609 abrogated the antiviral effect of RA, implying that RA depends on CAMK2 to suppress virus.
Overall, the authors use two in vitro cell line systems in the MHV model to clarify the mechanism through which RA is antiviral, acting through CAMKK and IRF3 to activate antiviral IFITs. As CAMKKs have a major role in neuronal signaling (with CAMK2 mRNA relocalization), and RA pathway hyperactivation (eg, vitamin A toxicity, isotretinoin (a very widely prescribed medication), and type I interferon adverse events) can be associated with fatal psychiatric outcomes, with limited mechanism to guide risk stratification for same, it seems important to further explore the link between this antiviral response pathway and neurological outcomes as the authors may have uncovered a crucial link helping start to clarify why certain patients have adverse neuro/psych outcomes when these pathways are upregulated.
Overall I think the manuscript is near ready for publication with recommendation to address the following minor comments:
The authors should justify why they chose the cell types they did, and explain why they used fibroblasts for some experiments and macrophages for others, and whether they looked at hepatocytes which would seem to be an important cell to explore for a hepatitis virus. If there is no specific scientific explanation, it should be stated that this was done for convenience.
The authors should provide the RA dose survival curve for RAW 264.7 cells as they do for the L929 cells.
The western blot graph (Fig 4A) should ideally be presented in the order of the gel (ie, IFNb at far R) for easier interpretation.
Comment on the importance of CAMKK in depression/psychiatric disease, the known risk for suicide/depression in vitamin A toxicity, isotretinoin, and IFN treatment, and potential importance of further clarifying how RA impacts this pathway in appropriate neuronal/psychiatric models.
Author Response
We value Reviewer #1’s positive remarks and constructive feedback. Additional information has been added to the manuscript to increase its scientific rigor (i.e., RA dose survival curve for RAW cells), and figure 4 has been updated to improve legibility. Likewise, the discussion section has been lengthened to describe the possible association between RA and CaMKK in the context of psychiatric disease (i.e., depression). Although not the focus of our manuscript, there is ample evidence highlighting a connection between RA and CaM signaling proteins to depression, separately. The opening paragraph of the results section was also expanded to detail the rationale behind the study’s cell-line choice.
Reviewer 2 Report
Comments and Suggestions for Authors
In this interesting manuscript the authors establish that all-trans retinoic acid (RA) is an inhibitor of MHV replication in cell cultures, and its inhibitory effect is dependent upon an intact IRF-3 gene. RNAseq analysis demonstrated that CaM kinase kinase 1 (CaMKK1) was upregulated by RA treatment of the RAW macrophage cell line and that an inhibitor of CAMKK1 enzymatic activity also blocked RA’s inhibition of MHV replication. RA treatment of cells has been shown to result in the inhibition of other viruses, and this effect has generally been associated with increased induction of interferon-beta. The effect of RA treatment on MHV replication is rather modest, approximately a 3.5-fold decrease with very high levels of RA. This contrasts with the 100-fold reduction demonstrated on mumps virus replication at a much lower dose of RA. This somewhat detracts from my enthusiasm for the manuscript. However, the observations regarding CAMMK1 are interesting and to my knowledge novel. The manuscript could be improved in a number of ways enumerated in the specific Comments below.
1.line 33-37. AlthoughSARS-CoV-2 clearly caused a pandemic, it is really stretching the definition of pandemic to call the SARS-CoV outbreak of 2002-3 a pandemic and the sporadic zoonotic cases of MERS-CoV infection certainly don’t meet that definition. These two sentences should rewritten to be more precise.
Lies 42-43. This sentence is simply not true as written. Clearence of MHV infection requires intact adaptive immune responses in addition to intact innate immunity. This needs to be rewritten to reflect that.
Lines 58-59. Paxlovid has also been approved by the FDA (November 2023) for treatment of mild-moderate COVID-19. This sentence needs to be updated.
Lines 68-70. Reference 27 demonstrates that the RA receptor agonist used, AM580, likely inhibited MERS-CoV replication through its effect on SREB and lipid synthesis, rather than its effect on interferon induction. This sentence does not make this at all clear.
Lines 183-199, Western blot Analysis. The authors fail to state what antibody they used for western blot analysis, the source of the antibody, and the dilution they used for the western blot analysis.
Lines 338-340 and figure 2. By my calculation the decrease in MHV replication is approximately 3.5-fold. Although this is statistically significant, its biologic significance is arguable since virus growth is exponential. It would be more precise to state the fold-decrease in viral yield instead of leaving it to the reader to calculate from the presented data in place of stating that the inhibition was significant. To assess its biological significance the authors would need to demonstrate an effect in mice.
Figures S2 and S3. This is the most novel part of the manuscript to my mind. The authors should consider including these as additional figures rather than putting it in the supplementary data. The figures could also be improved by labeling the CAMKK1 data points in the figure.
Author Response
We appreciate Reviewer #2’s positive remarks and constructive feedback. Unclear sentences in the Introduction and Methods section were rephrased to provide a more accurate background. For example, the role of the innate and adaptive immune system during viral infection was described in greater detail. Also, the missing information in the western plot protocol was added (e.g., antibody vendor, dilution, etc.). The supplemental RNA-sequencing figures (i.e., S2 and S3) were made into main figures (i.e., Fig6, Fig7) and CaMKK1 expression was labeled.
Reviewer 3 Report
Comments and Suggestions for Authors
This study confirmed the activated vitamin A or all-trans retinoid acid (RA) can inhibit coronavirus replication and its mechanism therein. The article was well written and can be published after the authors address the following minor comments.
1. Line 463: There is the narration "our previous experiments demonstrate that RA suppressed MHV replication in 463 mouse L929, WT, and IRF3-KO cell lines". However, their experiments only showed RA suppressed MHV replication in WT cells and in line 429, there is the statement "the IRF3-KO exhibited no benefit from RA treatment following MHV infection".
2. Line 474"Duo to the absence of IRF3, IRF9 becomes the main activator of the type 1 IFN response in IRF3-KO cells". How the authors know it is the IRF9 that play the role, not the other IRFs?
3. In author contributions, there is a mistake with the author name Zkixing K. Pan, should be Zhixing K, Pan.
Author Response
We agree with Reviewer #3’s feedback, and we’ve made the recommended corrections. The spelling of Dr. Pan’s name was fixed and the error in line 463 was resolved (i.e., we clarified that RA only has an effect on L929 and WT cells). Because other IRF may be playing a role in type I IFN induction, we rephrased line 474 to better reflect this fact.
Reviewer 4 Report
Comments and Suggestions for Authors
The authors explore the mechanism of retinoic acid (RA) on MHV infection. They find RA significantly attenuates MHV infection via induction of IRF3. The authors also identify CaMKK as a significant signaling player in this pathway and provide strong genetic evidence the antiviral effects of RA are mediated by induction of IRF3 and CaMKK. This work will be beneficial to the field and the results presented are sound. Minor comments are listed below.
Minor comments:
-Introduction lines 55-61. Paxlovid gained full US FDA approval in May 2023 while molnupiravir has emergency use authorization. Both of these are described in the source cited (18) and the introduction should be updated to reflect this.
-Results statistics. The authors annotate data with letters or stars for corresponding significances. The description for these are missing from some figure legends.
-Line 76 L929 fibroblast cell line is missing a source
-Figure 2. The data shown in figure 2A seems inconsistent with 2B. In particular, in 2B there appear to be significantly more plaques in the MHV + 100uM RA image than in the 10uM condition. This is not reflected in the quantitative data in 2A. Could the authors please address this and/or select a better representative plaque assay image?
-Figures S1, 3, 4, and 5 sizing is off, assuming this is a conversion issue from Word to the journal layout provided to reviewers but please ensure final figure formatting is consistent
-Line 700 More information for citation 18 is needed, insufficient as written.
Comments on the Quality of English LanguageGrammar and writing quality throughout the manuscript are fine overall, wording and word choice can be awkward at times.
Author Response
We appreciate the constructive feedback from Reviewer #4, and we have made the recommended updates. More information was added to the Materials Section to describe the origin of our L929 cells. Also, the introduction was amended to reflect the recent FDA approval for Paxlovid. Likewise, citation 18 was updated to include information on the associated author and publisher. The representative plaques shown in figure 2B were also replaced to better mirror the data illustrated in figure 2A.